# Association between preterm birth and economic and educational outcomes in adulthood: A population-based matched cohort study

Asma M. Ahmed[1], Eleanor Pullenayegum[1], Sarah D. McDonald[2], Marc Beltempo[3], Shahirose S. Premji[4], Jason D. Pole[5], Fabiana Bacchini[6], Prakesh S. Shah[7], Petros Pechlivanoglou[1]*

1 Child Health Evaluative Sciences, The Hospital for Sick Children, Toronto, ON, Canada, 2 Department of Obstetrics and Gynecology, Division of Maternal-Fetal Medicine, McMaster University, Hamilton, Ontario, Canada, 3 Department of Pediatrics, McGill University, Montreal, Quebec, Canada, 4 Faculty of Health Sciences, School of Nursing, Queen's University, Kingston, Ontario, Canada, 5 Centre for Health Services Research, The University of Queensland, Brisbane, Australia, 6 Canadian Premature Babies Foundation, Toronto, Ontario, Canada, 7 Department of Pediatrics, Mount Sinai Hospital, Toronto, Ontario, Canada

* petros.pechlivanoglou@sickkids.ca

**Data Availability Statement:** Due to Statistics Canada's access rules ensuring the confidentiality

## Abstract

### Background and objectives

Preterm birth (PTB) affects ~10% of births worldwide; however, most literature focused on short-term clinical outcomes, with much less focus on long-term socioeconomic outcomes after PTB. We examined associations between PTB and individuals' income, employment, and educational outcomes during early adulthood.

### Methods

We conducted a population-level matched cohort study including all live births in Canada between 1990 and 1996, followed until 2018. Outcomes included Employment income per year in 2018 CAD and employment between ages 18 and 28 years, postsecondary education enrollment (18–22 years), and maximum educational attainment at age 22–27 years. Mean differences and risk ratios (RR) and differences (RD) were estimated using generalized estimating equation regression models for economic outcomes and multinomial logistic regression models for educational outcomes.

### Results

Of 2.4 million births, 7% were born preterm (0.3%, 0.6%, 0.8%, and 5.4% born extremely preterm (24–27 weeks), very preterm (28–31 weeks), moderately preterm (32–33 weeks), and late preterm (34–36 weeks) respectively). After matching on baseline characteristics (e.g., sex, province of birth, and parental demographics) and adjusting for age and period effects, preterm-born individuals, on average, had $958 CAD less employment income per year (95% CI: $854-$1062), 6% lower income per year, than term-born individuals, and

of personal information collected under the Statistics Act, the linked data are only available through the Statistics Canada's Research Data Centres (RDCs) that form part of the Canadian Research Data Centre Network. Researchers with an approved project can access the RDCs after undergoing a security clearance and becoming deemed employees of Statistics Canada. Detailed information about the RDC application process and guidelines can be found at www.statcan.gc.ca/eng/rdc/process The data utilized in this study were accessed via the RDC network in accordance with Statistics Canada's standard application, approval, vetting, and disclosure policies for RDC access.

**Funding:** This study was supported by the Canadian Institutes of Health Research (grant# 438541) and Statistics Canada. AA received a postdoctoral fellowship from the Data Sciences Institute at the University of Toronto. The funders had no role in study design, data collection and analysis, decision to publish, or preparation of the manuscript.

**Competing interests:** The authors have declared that no competing interests exist.

were 2.13% less likely to be employed (1.98–2.29%). PTB was also negatively associated with university enrollment (RR 0.93 (0.91–0.94)) and graduation with a university degree (RR 0.95 (0.94–0.97)). Mean income differences for those born 24–27 weeks were -$5,463 CAD per year (17% lower), and adjusted RR were 0.55 for university enrollment and graduation.

## Conclusion

In this population-based study, preterm birth was associated with lower economic and educational achievements at least until the late twenties. The associations were stronger with decreasing GA at birth. Policymakers, clinicians, and parents should be aware that the socioeconomic impact of PTB is not limited to the early neonatal period but extends into adulthood.

## Introduction

Preterm birth (i.e., birth before 37 weeks of gestation, PTB) affects ~10% of births worldwide, and it accounts for almost a fifth of deaths in children < 5 years [1,2]. Preterm-born individuals, especially those born before 28 weeks of gestation, are vulnerable to numerous short and long-term morbidities, including neonatal complications and adverse neurodevelopmental outcomes [3–7]. Besides the direct medical expenses due to increased use of healthcare services, some children born preterm require special education and social services (~60% of those born <28 weeks) [8–11]. Families also face additional costs due to lifestyle and work adjustments [8,10,11]. For instance, parents of preterm infants often need extended leave from work, resulting in lost income and career setbacks. Families may incur higher childcare costs for specialized care and need home modifications for medical equipment or accessibility. These economic and lifestyle impacts can affect the family's financial stability and quality of life, potentially harming the child's long-term socioeconomic outcomes [8,10,11].

With more than 95% of preterm infants surviving to adulthood [12], studying long-term socioeconomic outcomes after PTB is important, as variations in these outcomes would have noticeable effects at the population level [13]. Most literature focuses on short-term clinical outcomes after PTB, whereas the association between PTB and economic and educational outcomes in adulthood is less studied [14]. Most previous studies have found negative associations between PTB and markers of adulthood wealth (educational attainment, income, and employment) [15–25], but some have shown no associations with socioeconomic outcomes during adulthood [26–28]. These differences could be attributed to differences in study design (prospective [15,20–22,26–28] vs registry-based [16–19,23–25]), outcome measures (self-reported [15,20–22,26–28] vs data linkage [16–19,23–25]), and study population (e.g., Europe [16–19,21,23,24,28] vs North America [15,20,22,25–27]). These studies were faced with methodological limitations [15,20,21,26–28]. Most studies have longitudinally tracked hospital-based cohorts of PTB survivors, along with term controls, and thus were vulnerable to selection bias due to differential loss of participants during follow-up [15,20,21,23,26–28]. The small sample size of most studies (n < 300) limits statistical power to detect small-to-moderate associations [15,20–22,26–28]. Further, most studies were restricted to those with gestational age (GA)< 28–32 weeks but did not consider the whole spectrum of PTB [15,20,21,26–28]. The few population-based studies on the educational and economic outcomes after PTB were conducted in

Scandinavia [16–19,24], where more inclusive educational and economic opportunities might be available, making these results less generalizable to different settings [14].

Utilizing a novel linkage of health, educational, and income tax data, we conducted a population-based matched cohort study in Canada to identify association between PTB and individuals' income, employment, and educational outcomes during adulthood.

## Materials and methods

### Settings

We created a longitudinal, population-based cohort study using the Social Data Linkage Environment (SDLE)–a secure environment for linking databases at Statistics Canada– to link datasets containing social, economic, and health data over time. Privacy and confidentiality were ensured during the linkage process and the subsequent use of coded linked data. Only authorized individuals accessed the coded (de-identified) data through a secure server at Statistics Canada. The Canadian Vital Statistics - Birth database (VSB) includes birth registration records for all live births in Canada, collected from all provincial and territorial vital statistics registries. The VSB file was linked to income tax data, postsecondary educational data, and mortality data. S1 Table presents further details on these databases. This study was approved by the Hospital for Sick Children Research Ethics Board, York University's Ethics Review Board, and Statistics Canada. As this study used de-identified data, individual-level informed consent was not required.

We included all live births in Canada between January 1, 1990, and December 31, 1996, identified from the VSB file. After identifying 2,729,400 eligible live births, we excluded 30,380 (1%) births with missing data on baseline covariates (GA, sex, birth plurality, and/or maternal age and parity), and 4,600 births with implausible birth weight and GA combination (birth weight for GA z score >4 SD above or below the mean). We further excluded births with GA <24 weeks' (because of the underreporting of neonatal deaths at early GA in Canada) [29,30] or >41 weeks' gestation (n = 2,690 and 105,270 respectively) given they were not part of the population of interest. We also excluded 149,260 individuals who did not have any income tax records during the follow-up period. For educational outcomes, we included births occurring from 1991 to 1996 because we only had complete information about postsecondary education for those born in 1991 onwards (see also Fig 1).

### Measures

**Exposure.** GA at birth, in completed weeks, identified from VSB was categorized as a binary preterm variable (24–36 weeks vs. 37–41 weeks) and a multicategory variable: extremely preterm (GA, 24–27 weeks), very preterm (28–31 weeks), moderately preterm (32–33 weeks), late preterm (34–36 weeks), and full-term births (37–41 weeks, the reference category). Information about gestational age in these data is obtained from the attending physician or the mother [31].

### Outcomes

**Economic outcomes.** The study cohort was followed until December 31, 2018, for a minimum of 22 years and a maximum of 28 years to ascertain the following outcomes:

*Employment income*: Employment income per year after an individual turns 18 years and until the end of follow-up was obtained from Statistics Canada income tax family files (T1FF). The T1FF records information on the personal income of individuals who file taxes in Canada (up to 8 months after the filing deadline of April 30[th]). We defined the total employment income

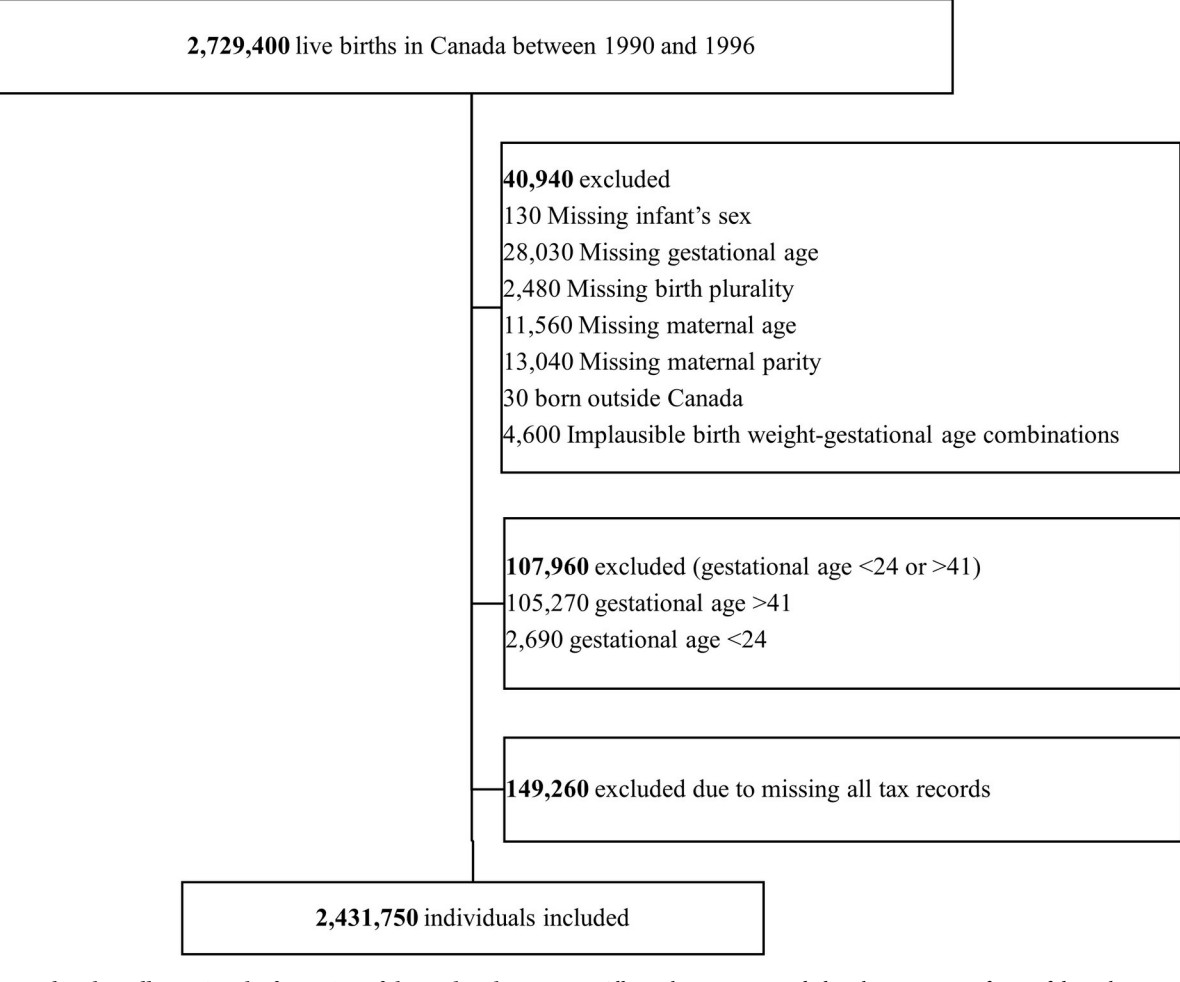

**Fig 1. Flowchart illustrating the formation of the study cohorts.** Note: All numbers were rounded to the nearest ten for confidentiality reasons.

as the annual individual pre-tax sum of paid employment income (wages, salaries, and commissions), net-self-employment income, and employment insurance income. If individuals filed taxes after the age of 18 years, we assumed zero income between age 18 and when they filed their first tax return. Employment income was adjusted to 2018 Canadian Dollars using the national-level Consumer Price Index to obtain a consistent measure of income over time [32].

*Employment*: We ascertained whether an individual, age >18 years, has reported any employment income during a calendar year (yes/no) from tax records.

**Educational outcomes.**   For educational outcomes, we used data from the Post-secondary Student Information System (PSIS)–a national database that collects information about postsecondary enrollment and graduation from public post-secondary institutions across Canada–covering calendar years between 2009 and 2018:

*Post-secondary Education Enrollment*: We determined individuals' enrollment in postsecondary education at age 18–22 years and categorized them according to the highest level of their program of enrollment as not enrolled, enrolled in college, or enrolled in a university program. We chose to restrict to age 18–22 years because most individuals are expected to enroll in a postsecondary program within this age range.

*Postsecondary Education Attainment*: We ascertained individuals' highest level of educational attainment by the end of follow-up (maximum age range 22–27 years) and categorized as a multinomial outcome: "did not graduate from any postsecondary education" (the reference) or "graduated with a non-university degree (International Standard Classification of Education (ISCED) level: 3–5)", "university degree (ISCED level: 6)", or "postgraduate degree/ medical professional training (ISCED level: 7–8, 96)".

## Baseline (matching) covariates

Baseline covariates included individual's sex, birth plurality, province and year of birth, and parental demographics at birth, including maternal marital status, parental age, parental place of birth [33], and maternal parity (details in S1 Table).

## Statistical analysis

We compared baseline maternal and child characteristics using standardized mean differences (SMD), with SMD >10% indicative of important differences between groups [34]. We used coarsened exact matching to reduce imbalance and potential confounding [35]. We created a set of strata representing unique combinations of covariate categories, as presented in Table 1. Any stratum with no reported preterm-born individuals or term-born controls was excluded [35]. All available term-born controls were selected for matching, and weights were generated so that the covariate distribution for controls matched that of the preterm-born individuals. We created separate matched cohorts for GA categories by matching individuals in each category to term-born controls.

For economic outcomes, the unmatched and matched cohorts were used to fit Generalizing Estimating Equation (GEE) linear regression models to estimate mean differences in employment income between groups, accounting for repeated measures and unequal follow-up time [36]. As income data tend to be skewed, we also examined differences in mean income on the log scale [37]. We used GEE- log-binomial regression to estimate risk ratios (RR) and differences (RD) for employment. We accounted for age and period effects by adjusting GEE models for age and calendar year modeled flexibly using restricted cubic splines [38,39]. Of all income tax records available in the data (from individuals who filed at least one income tax >18 years), approximately 6% of income tax records were missing, and we used multiple imputations by chained equations to impute income in those years (n = 5 imputations) [40]. We then estimated models across the five imputed datasets and pooled the results. For educational outcomes, we calculated RR using multinomial logistic regressions, accounting for clustering by mother using clustered variance estimates. Descriptive analyses, matching, and multiple imputations were conducted in R (version 4.2.1; The R Foundation). Stata version 16.1 (StataCorp, College Station, TX, USA) was used for GEE and multinomial regressions.

## Secondary analyses

We stratified all analyses by birth year and age categories. In the main analyses, we excluded all years after an individual's death. We conducted a sensitivity analysis where individuals who died during the follow-up were assigned zero income or were classified as unemployed or not enrolled/graduated to prevent selection bias introduced from differential losses to follow-up. Due to the substantial zero-mass of income data, we conducted a sensitivity analysis after excluding those with zero employment income. To account for confounding by family socioeconomic status, we used a subsample that was linked to maternal tax records at the time of birth (n = 1,625,480 births) and further matched on quintiles of maternal family income at or prior to birth (based on the average maternal family income during the two years before birth)

**Table 1. Characteristics of preterm and term-born individuals in the unmatched and matched Cohorts, Canada, 1983–1996 births.**

| Characteristics | Unmatched Cohort (n = 2.4 million), No. (%) | | | Matched Cohort (n = 2.1 million), No. (%) [weighted] | |
|---|---|---|---|---|---|
| | Term (n = 2,259,060) | Preterm (n = 172,690) | SMD | Term (n = 1,901,460) | Preterm (n = 157,500) |
| Individual's sex | | | 0.07 | | |
| Female | 1,109,170 (49.1) | 79,230 (45.9) | | 868,910 (45.7) | 71,970 (45.7) |
| Male | 1,149,890 (51.9) | 93,470 (54.1) | | 1,032,550 (54.3) | 85,530 (54.3) |
| Birth plurality | | | 0.50 | | |
| Singleton | 2,232,930 (98.8) | 148,810 (86.2) | | 1,720,930 (90.5) | 142,550 (90.5) |
| Multiple | 26,130 (1.2) | 23,880 (13.8) | | 180,540 (9.5) | 14,950 (9.5) |
| Maternal parity | | | 0.10 | | |
| 0 | 963,480 (42.6) | 79,200 (45.9) | | 904,170 (47.6) | 74,890 (47.6) |
| 1 | 812,090 (35.9) | 54,830 (31.7) | | 609,290 (32) | 50,470 (32) |
| 2 | 333,710 (14.8) | 24,800 (14.4) | | 262,110 (13.8) | 21,710 (13.8) |
| 3 | 100,560 (4.5) | 8,810 (5.1) | | 83,220 (4.4) | 6,890 (4.4) |
| >4 | 49,230 (2.2) | 5,060 (2.9) | | 42,680 (2.2) | 3,540 (2.2) |
| Maternal age | | | 0.10 | | |
| <20 years | 129,490 (5.7) | 12,150 (7.0) | | 135,120 (7.1) | 11,190 (7.1) |
| 20–24 years | 432,630 (19.2) | 33,730 (19.5) | | 375,640 (19.8) | 31,120 (19.8) |
| 25–29 years | 802,860 (35.5) | 56,310 (32.6) | | 635,640 (33.4) | 52,650 (33.4) |
| 30–34 years | 651,980 (28.9) | 48,040 (27.8) | | 531,110 (27.9) | 43,990 (27.9) |
| 35–39 years | 213,500 (9.5) | 19,300 (11.2) | | 198,610 (10.4) | 16,450 (10.4) |
| > 40 years | 28,600 (1.3) | 3,160 (1.8) | | 25,340 (1.3) | 2,100 (1.3) |
| Paternal age | | | 0.11 | | |
| <25 years | 245,940 (10.9) | 20,540 (11.9) | | 226,000 (11.9) | 18,720 (11.9) |
| 25–29 years | 624,360 (27.6) | 44,980 (26) | | 507,240 (26.7) | 42,020 (26.7) |
| 30–34 years | 735,760 (32.6) | 51,770 (30) | | 584,000 (30.7) | 48,370 (30.7) |
| 35–39 years | 355,840 (15.8) | 27,240 (15.8) | | 294,600 (15.5) | 24,400 (15.5) |
| >40 years | 149,200 (6.6) | 13,330 (7.7) | | 130,920 (6.9) | 10,840 (6.9) |
| Missing | 147,960 (6.5) | 14,820 (8.6) | | 158,710 (8.3) | 13,150 (8.3) |
| Maternal place of birth | | | 0.15 | | |
| Africa | 15,010 (0.7) | 1,390 (0.8) | | 10,350 (0.5) | 860 (0.5) |
| Asia | 112,720 (5.0) | 11,270 (6.5) | | 116,910 (6.1) | 9,680 (6.1) |
| Canada | 1,830,750 (81.0) | 131,400 (76.1) | | 1,490,460 (78.4) | 123,460 (78.4) |
| Central and South America | 32,300 (1.4) | 3,370 (1.9) | | 29,040 (1.5) | 2,410 (1.5) |
| Europe | 112,330 (5.0) | 8,440 (4.9) | | 80,420 (4.2) | 6,660 (4.2) |
| North America excluding Canada | 26,970 (1.2) | 1,790 (1.0) | | 12,470 (0.7) | 1,030 (0.7) |
| Other | 128,990 (5.7) | 15,040 (8.7) | | 161,820 (8.5) | 13,400 (8.5) |
| Paternal place of birth | | | 0.16 | | |
| Africa | 17,650 (0.8) | 1,540 (0.9) | | 11,140 (0.6) | 920 (0.6) |
| Asia | 109,490 (4.8) | 10,640 (6.2) | | 111,430 (5.9) | 9,230 (5.9) |
| Canada | 1,679,970 (74.4) | 117,970 (68.3) | | 1,342,720 (70.6) | 111,220 (70.6) |
| Central and South America | 32,580 (1.4) | 3,220 (1.9) | | 27,980 (1.5) | 2,320 (1.5) |
| Europe | 128,490 (5.7) | 9,370 (5.4) | | 90,690 (4.8) | 7,510 (4.8) |
| North America excluding Canada | 21,470 (1) | 1,480 (0.9) | | 10,000 (0.5) | 830 (0.5) |
| Other | 269,400 (11.9) | 28,480 (16.5) | | 307,490 (16.2) | 25,470 (16.2) |
| Maternal marital status at birth | | | 0.14 | | |
| Married | 1,560,010 (69.1) | 109,230 (63.3) | | 1,224,190 (64.4) | 101,400 (64.4) |
| Other | 103,550 (4.6) | 12,420 (7.2) | | 126,790 (6.7) | 10,500 (6.7) |
| Missing | 41,770 (1.8) | 3,730 (2.2) | | 29,800 (1.6) | 2,470 (1.6) |
| Single | 553,730 (24.5) | 47,310 (27.4) | | 520,690 (27.4) | 43,130 (27.4) |
| Maternal place of residence at the time of giving birth | | | 0.11 | | |
| Alberta | 244,480 (10.8) | 18,300 (10.6) | | 195,300 (10.3) | 16,180 (10.3) |
| Atlantic Provinces | 165,250 (7.3) | 12,410 (7.2) | | 135,920 (7.1) | 11,260 (7.1) |
| British Columbia | 270,590 (12) | 18,100 (10.5) | | 189,800 (10) | 15,720 (10) |
| Manitoba | 98,290 (4.4) | 7,940 (4.6) | | 81,490 (4.3) | 6,750 (4.3) |
| Ontario | 854,880 (37.8) | 73,300 (42.4) | | 823,570 (43.3) | 68,220 (43.3) |
| Quebec | 528,470 (23.4) | 35,960 (20.8) | | 405,620 (21.3) | 33,600 (21.3) |
| Saskatchewan | 87,150 (3.9) | 6,040 (3.5) | | 63,950 (3.4) | 5,300 (3.4) |
| Yukon, Nunavut, and Northwest Territory | 9,940 (0.4) | 640 (0.4) | | 5,820 (0.3) | 480 (0.3) |

*(Continued)*

**Table 1.** (Continued)

| Characteristics | Unmatched Cohort (n = 2.4 million), No. (%) | | | Matched Cohort (n = 2.1 million), No. (%) [weighted] | |
|---|---|---|---|---|---|
| | Term (n = 2,259,060) | Preterm (n = 172,690) | SMD | Term (n = 1,901,460) | Preterm (n = 157,500) |
| Birth year | | | 0.08 | | |
| 1990 | 329,490 (14.6) | 22,430 (13) | | 250,370 (13.2) | 20,740 (13.2) |
| 1991 | 329,850 (14.6) | 23,380 (13.5) | | 258,540 (13.6) | 21,420 (13.6) |
| 1992 | 332,890 (14.7) | 23,940 (13.9) | | 265,640 (14) | 22,000 (14) |
| 1993 | 321,780 (14.2) | 24,690 (14.3) | | 272,330 (14.3) | 22,560 (14.3) |
| 1994 | 323,650 (14.3) | 25,590 (14.8) | | 280,230 (14.7) | 23,210 (14.7) |
| 1995 | 316,990 (14) | 26,520 (15.4) | | 290,300 (15.3) | 24,050 (15.3) |
| 1996 | 304,400 (13.5) | 26,150 (15.1) | | 284,060 (14.9) | 23,530 (14.9) |

Note: All numbers were rounded to the nearest ten for confidentiality reasons.

and place of residence (rural or urban) obtained from maternal tax files and compared results. We also conducted sensitivity analyses for economic outcomes among 1983–1996 births (n = 4,388,200). This cohort was not used for primary analysis because of data quality issues before 1990 (e.g., poor linkage of T1FF to Quebec birth registry (17.5% of births)).

## Results

We included 2,431,750 live births in the analysis. Excluded individuals with no tax record during the entire follow-up period (~6%) were more likely to be born preterm or as part of multiple births and be born to young (<20 years old), single, or multiparous mothers, foreign-born parents, or missing father's age (S2 Table).

### Descriptive results

**Unmatched cohort.** The rate of preterm birth in the final study population was ~7% (n = 172,690); 5.4% (n = 131,960), 0.8% (n = 19,870), 0.3% (n = 14,460), 0.2% (n = 6,410) born with a GA of 34–36, 32–33, 28–31, and 24–27 weeks, respectively. Individuals born preterm were more likely to be male, born in more recent years or multiple gestation births, and be born to mothers <20 or >40 years, mothers with a single or missing marital status, multiparous mothers, and foreign-born parents (Tables 1 and S3).

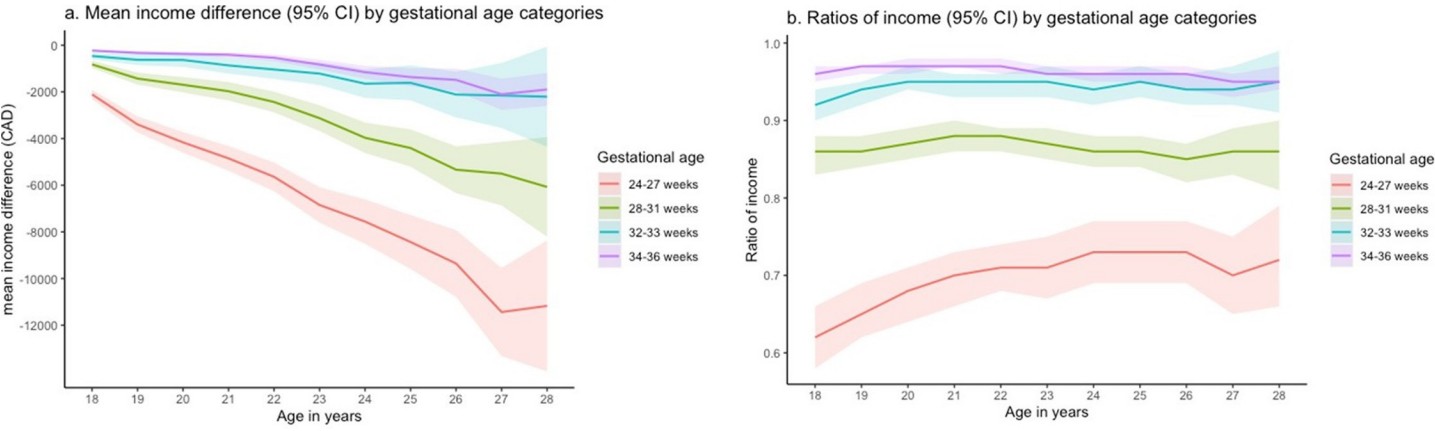

**Fig 2.** Differences (a) and ratios (b) of mean employment income according to gestational age categories by years of age in the matched cohorts.

Individuals were followed until the end of 2018 for a median of 25 years of age (IQR 23–27 years) to ascertain economic outcomes. The mean employment income per year at age 23–28 years was $30,600 CAD for those born at term and $28,400 CAD for those born preterm. At 23–28 years, 89% of those born at term were employed vs. 86% for those born preterm (S4 Table).

Among individuals born between 1991–1996 (n = 2,079,830 births), at age 18–22 years, 27% and 40% of those born at term were enrolled in a college or university respectively, compared to 28% and 37% of those born preterm. By the end of follow-up, the percentages of individuals who graduated with a non-university degree, a university degree, and a postgraduate degree were 20%, 21%, and 2% for term and 19%, 19%, and 2% for preterm-born individuals respectively (S4 Table).

**Matched cohort.** The matched cohort included 2,058,960 births (157,500 preterm and 1,901,460 term-born individuals). As expected, differences in the distribution of characteristics by PTB were eliminated in the matched cohort (all SMD<0.001) (Tables 1 and S3). Fig 2A and 2B illustrate the differences and ratios respectively, in mean employment income. Those born preterm had lower mean income, with larger differences as GA decreased and increasing gaps over time. Relative differences in income were also more pronounced in the low GA categories, but the differences were relatively stable over time. Fig 3 shows risk differences for employment by age according to gestational age categories. Larger differences were identified as the GA decreased but gaps became slightly smaller with age.

## Associations between PTB and economic outcomes

The mean employment income for individuals born preterm was lower than those born at term (mean differences per year: -$958 CAD (-$1,062, -$854), 6% lower, in the matched cohort, adjusting for age and period effects) (Table 2). Those born preterm were less likely to be employed (RR: 0.98 (0.98–0.98); RD -2.13% (-2.29%, -1.97%) in the matched cohort). These differences increased as GA decreased. For example, mean income difference per for those born 24–27 weeks was -$5,463 CAD (27% lower), and risk difference for employment was -13.54%.

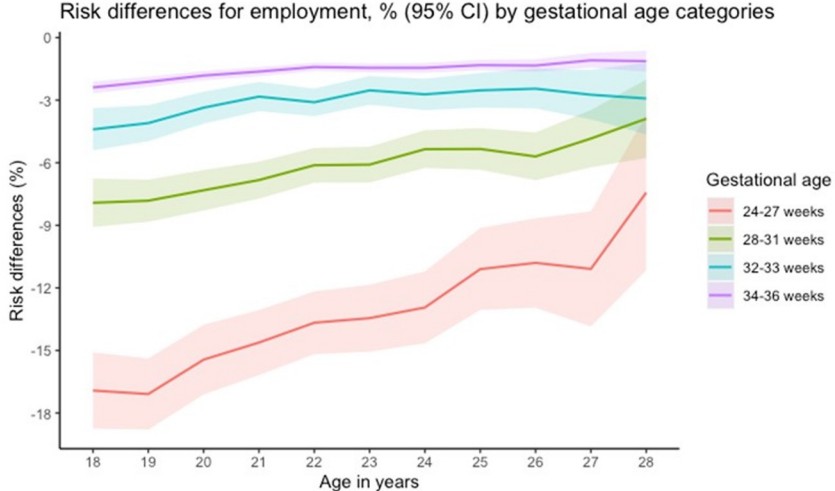

**Fig 3. Risk differences (%) for employment according to gestational age categories by years of age in the matched cohorts.**

**Table 2.** Associations between preterm birth and employment income (a) and employment (b) per year, at or after the age of 18 years for individuals born in 1990–1996 in Canada.

**(a) Employment income**

| | Mean income differences in CAD (95% CI) | | |
|---|---|---|---|
| Category | Unmatched cohort [a] | Matched cohort model 1 [b,c] | Matched cohort model 2 [c,d] |
| Preterm birth | | | |
| Preterm (<37 weeks) | -1608 (-1680, -1537) | -971 (-1077, -864) | -958 (-1062, -854) |
| Gestational age category | | | |
| Late preterm births (34–36 weeks) | -1331 (-1411, -1251) | -653 (-780, -527) | -648 (-772, -524) |
| Moderately preterm births (32–33 weeks) | -1511 (-1716, -1305) | -1046 (-1278, -814) | -1036 (-1261, -811) |
| Very preterm births (28–31 weeks) | -2894 (-3131, -2658) | -2547 (-2816, -2278) | -2518 (-2780, -2256) |
| Extremely preterm births (24–27 weeks) | -6295 (-6671, -5919) | -5531 (-5958, -5105) | -5463 (-5879, -5046) |
| | Ratios of income (95% CI) | | |
| | Unmatched cohort [a] | Matched cohort model 1 [b,c] | Matched cohort model 2 [c,d] |
| Preterm birth | | | |
| Preterm (<37 weeks) | 0.91 (0.91, 0.92) | 0.95 (0.94, 0.95) | 0.94 (0.93, 0.95) |
| Gestational age category | | | |
| Late preterm births (34–36 weeks) | 0.93 (0.93, 0.93) | 0.96 (0.96, 0.97) | 0.96 (0.94, 0.97) |
| Moderately preterm births (32–33 weeks) | 0.92 (0.91, 0.93) | 0.94 (0.93, 0.96) | 0.95 (0.93, 0.96) |
| Very preterm births (28–31 weeks) | 0.85 (0.83, 0.86) | 0.86 (0.85, 0.88) | 0.86 (0.84, 0.88) |
| Extremely preterm births (24–27 weeks) | 0.67 (0.65, 0.69) | 0.70 (0.68, 0.72) | 0.73 (0.70, 0.75) |

**(b) Employment**

| | Risk differences (%) for employment (95% CI) | | |
|---|---|---|---|
| | Unmatched cohort [a] | Matched cohort model 1 [b,c] | Matched cohort model 2 [c,d] |
| Preterm birth | | | |
| Preterm (<37 weeks) | -4.01 (-4.15, -3.87) | -2.62 (-2.78, -2.46) | -2.13 (-2.29, -1.97) |
| Gestational age category | | | |
| Late preterm births (34–36 weeks) | -3.09 (-3.25, -2.94) | -1.74 (-1.91, -1.57) | -1.37 (-1.54, -1.19) |
| Moderately preterm births (32–33 weeks) | -4.39 (-4.8, -3.97) | -3.27 (-3.74, -2.81) | -2.66 (-3.14, -2.19) |
| Very preterm births (28–31 weeks) | -8.05 (-8.60, -7.50) | -6.70 (-7.31, -6.09) | -5.86 (-6.47, -5.26) |
| Extremely preterm births (24–27 weeks) | -17.15 (-18.24, -16.06) | -14.54 (-15.72, -13.37) | -13.54 (-14.73, -12.35) |
| | Risk ratios for employment (95% CI) | | |
| | Unmatched cohort [a] | Matched cohort model 1 [b,c] | Matched cohort model 2 [c,d] |
| Preterm birth | | | |
| Preterm (<37 weeks) | 0.95 (0.95, 0.95) | 0.97 (0.97, 0.97) | 0.98 (0.98, 0.98) |
| Gestational age category | | | |
| Late preterm births (34–36 weeks) | 0.96 (0.96, 0.97) | 0.98 (0.98, 0.98) | 0.99 (0.98, 0.99) |
| Moderately preterm births (32–33 weeks) | 0.95 (0.94, 0.95) | 0.96 (0.96, 0.97) | 0.97 (0.97, 0.98) |
| Very preterm births (28–31 weeks) | 0.90 (0.90, 0.91) | 0.92 (0.91, 0.93) | 0.94 (0.93, 0.95) |
| Extremely preterm births (24–27 weeks) | 0.80 (0.78, 0.81) | 0.82 (0.81, 0.84) | 0.83 (0.82, 0.85) |

[a] n = 2,411,230 individuals; 19,342,590 person-year.

[b] Matched cohort model 1 used the matched sample.

[c] n = 2,041,850 for <37 category (16,416,650 person-years); n = 1,962,730 for 34–36 category (15,784,750 person-years); n = 1,320,350 for 32–33 category (10,682,710 person-years); n = 1,192,190 for 28–31 category (9,718,060 person-years); n = 842,890 for 24–27 category (6,888,950 person-years).

[d] Matched cohort model 2 used the matched sample and further adjusted for the calendar year and age modeled using restricted cubic splines.

## Associations between PTB and educational outcomes

PTB was negatively associated with college or university enrollment (17% less likely to enroll in a university program, RR 0.83 (0.81–0.84) in the matched cohort). Preterm-born individuals were also 16% less likely to graduate with a university degree (RR 0.84 (0.83, 0.86)). Estimates by GA categories showed an inverse association with GA. For example, RRs for the extremely preterm-born individuals (24–27 weeks) were 0.55 for university enrollment or graduation in the matched cohort (Table 3).

**Table 3.** Associations between preterm birth and (a) postsecondary education enrollment (age 18–22 years) and (b) postsecondary education attainment (age 22–27 years) for individuals born in 1991–1996 in Canada.

(a) Postsecondary education enrollment (reference category: did not enroll in any postsecondary education)

| | Risk ratios for postsecondary education enrollment (95% CI) | | | |
|---|---|---|---|---|
| | College | | University | |
| Category | Unmatched [a] | Matched [b] | Unmatched [a] | Matched [b] |
| Preterm birth | | | | |
| Preterm (<37 weeks) | 0.92 (0.9, 0.93) | 0.93 (0.91, 0.94) | 0.83 (0.82, 0.84) | 0.83 (0.81, 0.84) |
| Gestational age category | | | | |
| Late preterm births (34–36 weeks) | 0.93 (0.91, 0.94) | 0.93 (0.92, 0.95) | 0.86 (0.85, 0.88) | 0.85 (0.83, 0.86) |
| Moderately preterm births (32–33 weeks) | 0.91 (0.88, 0.95) | 0.93 (0.89, 0.97) | 0.80 (0.77, 0.83) | 0.81 (0.77, 0.84) |
| Very preterm births (28–31 weeks) | 0.86 (0.82, 0.90) | 0.92 (0.87, 0.97) | 0.69 (0.66, 0.72) | 0.72 (0.69, 0.76) |
| Extremely preterm births (24–27 weeks) | 0.82 (0.76, 0.88) | 0.88 (0.81, 0.96) | 0.53 (0.49, 0.57) | 0.55 (0.51, 0.60) |

[a] n = 2,058,980 individuals.

[b] n = 1,737,940 for <37 category; n = 1,670,620 for 34–36 category; n = 1,115,630 for 32–33 category; n = 997,620 for 28–31 category; n = 703,250 for 24–27 category.

(b) Postsecondary education attainment (reference category: did not graduate from any postsecondary education)

| | Risk ratios for postsecondary education attainment (95% CI) | | | | | |
|---|---|---|---|---|---|---|
| | Non-University | | University | | Postgraduate | |
| Category | Unmatched [a] | Matched [b] | Unmatched [a] | Matched [b] | Unmatched [a] | Matched [b] |
| Preterm birth | | | | | | |
| Preterm (<37 weeks) | 0.89 (0.88, 0.91) | 0.95 (0.94, 0.97) | 0.83 (0.82, 0.85) | 0.84 (0.83, 0.86) | 0.78 (0.75, 0.81) | 0.85 (0.82, 0.89) |
| Gestational age category | | | | | | |
| Late preterm (34–36) | 0.9 (0.89, 0.92) | 0.96 (0.94, 0.97) | 0.86 (0.85, 0.87) | 0.86 (0.85, 0.88) | 0.81 (0.78, 0.85) | 0.89 (0.85, 0.93) |
| Moderately preterm (32–33) | 0.88 (0.84, 0.92) | 0.95 (0.91, 1.00) | 0.82 (0.78, 0.85) | 0.84 (0.80, 0.88) | 0.78 (0.69, 0.87) | 0.87 (0.77, 0.98) |
| Very preterm (28–31) | 0.87 (0.83, 0.91) | 0.94 (0.89, 1.00) | 0.71 (0.67, 0.75) | 0.74 (0.70, 0.79) | 0.63 (0.54, 0.73) | 0.67 (0.57, 0.78) |
| Extremely preterm (24–27) | 0.82 (0.76, 0.89) | 0.94 (0.86, 1.03) | 0.55 (0.50, 0.60) | 0.55 (0.49, 0.61) | 0.40 (0.29, 0.54) | 0.46 (0.33, 0.64) |

[a] n = 2,063,480 individuals.

[b] n = 1,741,620 for <37 category; n = 1,674,090 for 34–36 category; n = 1,117,850 for 32–33 category; n = 999,560 for 28–31 category; n = 704,630 for 24–27 category.

## Secondary analyses

Analyses stratified by birth cohort showed that differences in economic and educational outcomes have remained relatively stable across birth cohorts (S5 and S6 Tables). For economic outcomes, analyses stratified by age group revealed that absolute differences in mean employment income by PTB (and GA categories) increased with increased age (mean differences by PTB -$621 CAD per year at age 18–22 and -$2,202 at age 26–28 years in the matched cohort); however, relative differences in income have remained stable (S5 Table). In analyses where deaths were assigned zero income or were classified as unemployed or not enrolled/graduated, differences by PTB became slightly larger, but as expected, differences were more pronounced in the extremely preterm group (24–27 weeks) (mean income differences -$9,280 CAD per year; RR 0.58 for unemployment; RR 0.28 for university enrollment; and RR 0.34 university graduation) (S7 and S8 Tables). For employment income as an outcome, analyses that excluded individuals reporting zero employment income showed smaller differences, especially in the extremely preterm birth subgroup (mean income differences -$4,287 CAD per

year; 17% lower income per year) (S9 Table). Accounting for family income at baseline, along with other baseline characteristics, had little impact on the results (annual mean income difference: -$721 CAD (-821, -622), 4% lower; RR for employment0.98 (0.98, 0.98); RR for university enrollment 0.86 (0.84, 0.88); RR for university graduation 0.89 (0.87, 0.90)) (S10 and S11 Tables). Analyses of economic outcomes among 1983–1996 births showed similar patterns as the main results (S12 Table).

## Discussion

In this population-based cohort of ~2.4 million individuals, the average employment income of preterm-born individuals was ~$1,000 CAD (6%) lower per year than those born at term. Those born preterm were also slightly less likely to be employed, to enroll in college or university programs, or to complete postsecondary education. These variations differed by GA, with minor differences in individuals born at 34–36 weeks and larger differences as GA decreased.

Some previous small prospective cohort studies of individuals born preterm found no differences in socioeconomic outcomes [26–28]. This may be due to the limitations of prospective designs, such as small sample sizes, reliance on self-reported socioeconomic data, and higher loss of follow-up among those with low socioeconomic status [14]. Similar to our findings, several registry-based, population-based studies have documented poorer educational and economic outcomes among those born preterm [16–19,24]. Two Swedish studies (1973–1979 births and 1973–2008 births) found lower educational levels, earnings, and unemployment among preterm-born individuals than term-born individuals [16,19]. Likewise, a Norwegian population-based study that followed individuals born in 1967–1983 until the end of 2003 found a negative association between GA at birth and levels of education and the risk of having low income or receiving social assistance [18]. The magnitude of associations in some of these studies, however, was smaller than ours, likely because they excluded individuals with medical disabilities [16,18]. We also focused on employment income (which is more sensitive to changes in labor market participation), while other studies have examined other forms of income, such as total income or income from social welfare [16,18].

Our results stratified by birth year showed similar differences across birth cohorts. Although cross-study comparisons are challenging due to variations in study designs and outcome measures, previous studies' results were generally consistent between older and more recent cohorts, suggesting persistent economic gaps despite advances in neonatal care and changes in societal structures [16–19,24].

We found that the risk of adverse socioeconomic outcomes increases with decreased GA. While only a few studies have considered the whole spectrum of GA, they similarly found an inverse association between GA and socioeconomic outcomes [18,19,24]. For example, RRs in Moster's study for the 23–27, 28–30, 31–33, and 34–36 GA categories were approximately 0.8, 0.9, 1.0, and 1.0 for completing a university degree and 0.8, 0.9, 0.9, and 1.0 for high income respectively [18]. The Danish study that used similar categorization of educational attainment using ISCED levels also reported lower educational levels among preterm individuals, with stronger associations as GA decreased (RR 0.21, 0.45, 0.67, and 0.84 for 22–27, 28–31, 33, and 36 weeks of gestation) [24].

Several potential mechanisms could explain adverse socioeconomic outcomes among individuals born preterm. The third trimester of pregnancy is a period of rapid brain growth, and therefore, PTB might impair brain maturation and development [41,42], a hypothesis that is supported by several studies in the literature [43,44]. Lower GA at birth is also associated with several health and neurodevelopmental challenges [3–7], which could affect individuals' academic achievements. Furthermore, PTB more commonly occurs in socially disadvantaged

families [45], and PTB might further lower the family's socioeconomic position due to increased economic stress on families [11,25,46], thereby lowering access to supportive services.

## Strengths & limitations

Strengths of this study include the large nationally representative data, the long duration of follow-up (22–28 years), and the use of several socioeconomic outcomes ascertained from government administrative databases (national tax database and postsecondary education registry), eliminating self-reporting bias. The large sample size allowed for sufficient statistical power for analyses stratified by standard PTB sub-categories.

Our study also has several limitations. First, our study is based on administrative data and is thus vulnerable to measurement errors. Educational outcomes were ascertained from a national database that only includes information about postsecondary enrollment and graduation from public Canadian institutions; therefore, those who sought postsecondary education at private institutions or abroad would be misclassified as not enrolled/graduated [47]. For economic outcomes, we only examined employment income as recorded on tax returns, but we did not account for income not reported when filing taxes, such as cash payments. Despite several incentives for low-income individuals to file their tax returns (e.g., social benefits), some may not meet the administrative deadline (within 8 months from the April 30th filing deadline), resulting in missing income data. Measurement errors in GA are also likely, particularly since we relied on birth certificate data that are mother- or physician-reported; however, we excluded births with an implausible combination of GA and birth weight to minimize misclassification.

Not all individuals eligible for inclusion were found in the tax database, and those born preterm were more likely to have all their tax records missing (~6% of individuals born at term vs. 8–13% for those born preterm). Underreporting of perinatal deaths is a possibility, which has been reported in Canada, especially in earlier births [29,30]. Individuals with missing tax records could also have moved outside of Canada, as they were more likely to have foreign-born parents. Some of these individuals might also have zero income due to an illness or disability that precluded them from participating in the workforce. Excluding these individuals would likely underestimate the associations between PTB and the study outcomes.

Although we accounted for differences in baseline characteristics between preterm and term-born births using matching, residual confounding cannot be ruled out given the observational nature of the study. For example, we did not adjust for parental social factors such as education, occupation, lifestyle factors, and maternal morbidities and mental health conditions, which have been linked to both the risk of PTB and worse socioeconomic outcomes [45,48–50]. Findings on individuals born between 1990 and 1996 may not be generalizable to recent birth cohorts, as survival and treatment of preterm-born children have changed during the study period, particularly for those born <28 weeks' gestation.

## Implications

Policymaker, care providers, and parents should be aware that the socioeconomic effects of PTB is not limited to the early neonatal period but may extend into adulthood. These findings should be seen as a call to action rather than discouragement, emphasizing that all individuals, regardless of their present or future economic contributions, receive the care and support they need. By understanding the potential challenges associated with PTB, these results could inform policies that promote access to educational and developmental intervention programs in early life to improve long-term outcomes. These findings could also help parents make long-term plans to navigate the potential financial and educational hardships of their children.

In addition, policies that support ongoing academic and vocational training and access to opportunities could help reduce these disparities. For healthcare providers, providing comprehensive care, regular monitoring and developmental surveillance, transition planning, coordination of care, and ongoing support for families, might help improve outcomes after PTB. Future studies that focus on identifying the role of modifiable factors that mediate or moderate these associations is needed to inform effective intervention strategies. In addition, studies should consider a longer follow-up to observe if these associations extend to mid-late adulthood and beyond. Future work should also elucidate potential mediating variables for the observed associations (e.g., developmental disabilities). Additionally, it should highlight long-term impacts of PTB on social functioning and family relationships.

## Conclusion

In this population-based matched cohort study, preterm birth was associated with lower economic and educational achievements at least until the late twenties. The association was stronger with decreasing GA. Increasing awareness and investments in early life intervention to improve the economic and educational outcomes after PTB are needed.

## Supporting information

**S1 Table. Description of different datasets at Statistics Canada used in the study and the relevant study variables.**
(DOCX)

**S2 Table. Characteristics of individuals included vs excluded from the study population [N (%)].**
(DOCX)

**S3 Table.** Characteristics of the unmatched (a) and matched cohorts (b) according to gestational age category [N (%).
(DOCX)

**S4 Table. Descriptive statistics for study outcomes by preterm birth (and gestational age categories) in the unmatched cohorts.**
(DOCX)

**S5 Table.** Associations between preterm birth and employment income and employment per year, at or after the age of 18 years for individuals born in 1990–1996 in Canada, stratified by age group (a) and birth cohort (b).
(DOCX)

**S6 Table.** Associations between preterm birth and (a) postsecondary education enrollment (age 18–22 years) and (b) attainment (age 22–27 years) for individuals born in 1991–1996 in Canada, stratified by birth cohort.
(DOCX)

**S7 Table. Associations between preterm birth and employment income and employment per year, at or after the age of 18 years for individuals born in 1990–1996 in Canada when individuals who died were assigned zero income or unemployed.**
(DOCX)

**S8 Table.** Associations between preterm birth and (a) postsecondary education enrollment (age 18–22 years) and (b) attainment (age 22–27 years) for individuals born in 1991–1996 in Canada when individuals who died were categorized as not enrolled/graduated from

postsecondary education.
(DOCX)

**S9 Table. Associations between preterm birth and employment income per year at or after the age of 18 years for individuals born in 1990–1996 in Canada, excluding individuals with zero employment income.**
(DOCX)

**S10 Table. Associations between preterm birth and employment income and employment per year, at or after the age of 18 years for individuals born in 1990–1996 in Canada in the subsample linked to maternal tax with and without matching on maternal income and rural residence.**
(DOCX)

**S11 Table.** Associations between preterm birth and (a) postsecondary education enrollment (age 18–22 years) and (b) attainment (age 22–27 years) for individuals born in 1991–1996 in Canada in the subsample linked to maternal tax with and without matching on maternal income and rural residence.
(DOCX)

**S12 Table. Associations between preterm birth and employment income and employment per year, at or after the age of 18 years for individuals born in 1983–1996 in Canada.**
(DOCX)

## Author Contributions

**Conceptualization:** Asma M. Ahmed, Eleanor Pullenayegum, Sarah D. McDonald, Marc Beltempo, Shahirose S. Premji, Jason D. Pole, Fabiana Bacchini, Prakesh S. Shah, Petros Pechlivanoglou.

**Formal analysis:** Asma M. Ahmed.

**Funding acquisition:** Eleanor Pullenayegum, Sarah D. McDonald, Marc Beltempo, Shahirose S. Premji, Jason D. Pole, Fabiana Bacchini, Prakesh S. Shah, Petros Pechlivanoglou.

**Methodology:** Asma M. Ahmed, Eleanor Pullenayegum, Petros Pechlivanoglou.

**Software:** Asma M. Ahmed.

**Supervision:** Eleanor Pullenayegum, Prakesh S. Shah, Petros Pechlivanoglou.

**Writing – original draft:** Asma M. Ahmed.

**Writing – review & editing:** Asma M. Ahmed, Eleanor Pullenayegum, Sarah D. McDonald, Marc Beltempo, Shahirose S. Premji, Jason D. Pole, Fabiana Bacchini, Prakesh S. Shah, Petros Pechlivanoglou.

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
