## [Decision Letter · Decision Letter 0]

2 Jun 2024

PONE-D-23-38655Association between preterm birth and economic and educational outcomes in adulthood: A population-based matched cohort studyPLOS ONE

Dear Dr. Pechlivanoglou,

Thank you for submitting your manuscript to PLOS ONE. After careful consideration, we feel that it has merit but does not fully meet PLOS ONE’s publication criteria as it currently stands. Therefore, we invite you to submit a revised version of the manuscript that addresses the points raised during the review process.

 Additionally, thank you for your patience with this review process. Overall your manuscript has received positive comments. Further, the reviewers have provided in-depth and constructive feedback that will truly strengthen your manuscript if well considered. Please respond to all of the reviewer's suggestions. I would like to emphasize the need to address why family of origin SES was not included in the models. If it is available, including how this adjustment impacts the models would be helpful even if just presented in supplemental analyses. Again, thank you for your time.

We look forward to receiving your revised manuscript.

Kind regards,

Quetzal A. Class, PhD

Academic Editor

PLOS ONE

 [This study was supported by the Canadian Institutes of Health Research (grant# 438541) and Statistics Canada. AA received a postdoctoral fellowship from the Data Sciences Institute at the University of Toronto].  

Reviewers' comments:

Reviewer's Responses to Questions

**Comments to the Author**

1. Is the manuscript technically sound, and do the data support the conclusions?

Reviewer #1: Yes

Reviewer #2: Partly

2. Has the statistical analysis been performed appropriately and rigorously? 

Reviewer #1: Yes

Reviewer #2: Yes

3. Have the authors made all data underlying the findings in their manuscript fully available?

Reviewer #1: Yes

Reviewer #2: Yes

4. Is the manuscript presented in an intelligible fashion and written in standard English?

Reviewer #1: Yes

Reviewer #2: Yes

5. Review Comments to the Author

Reviewer #1: To the authors

In this manuscript, authors conducted a population-level matched cohort study using ~2.4 million individuals in Canada between 1990 and 1996, followed until 2018. By using the following parameters: employment income per year in 2018 CAD, employment between ages 18 and 28 years, postsecondary education enrollment (18-22 years), and maximum educational attainment at age 22-27 years, they found (1) that the average employment income of preterm-born individuals was ~$1,000 CAD (6%) lower per year than those born at term, (2) that those born preterm were also slightly less likely to be employed and less likely to enroll in college or university programs or complete postsecondary education, and (3) that these variations differed by GA, with minor differences in individuals born at 34-36 weeks and larger differences as GA decreased.

This is a great study with large sample size which allowed for sufficient statistical power for analyses stratified by standard PTB sub-categories. Furthermore, this study include the large nationally representative data, the long duration of follow-up (22-28 years), and the use of several socioeconomic outcomes ascertained from government administrative databases.

Major problems

No major problems

Comments.

Our goal as a neonatologist is no longer just improved survival rate, but also intact survival, and therefore, we always think about whether PTB can become independent as a human being and contribute to society. This study results nicely demonstrated these points, and I hope the governments will promote access to educational and developmental intervention programs in early life to improve long-term outcome, with these outcomes.

However, what we must be careful about is that lives should not be chosen just because they do not contribute to the economy, as the authors mentioned in Discussion. So, I would like you intensively to emphasize this point.

Also, as times have changed, I think there are various forms of employment and education that did not exist in the past. Moreover, this study did not include parameters solely from the family's perspective, and did not evaluate family happiness, which cannot be evaluated from an economic perspective. Of course, I know this is a scientific paper, and the content should not be spiritual or conceptual, but I think it should never go in the direction of discouraging neonatal medicine. I wanted you to keep in mind that point.

Reviewer #2: Thank you for the opportunity to review this manuscript titled “Association between preterm birth and economic and educational outcomes in adulthood: A population-based matched cohort study”. This study offers valuable insights into the long-term effects of preterm birth, conducted using tax data for a large, population-based cohort of approximately 2.4 million individuals born in Canada between 1990 and 1996. The results presented indicate disadvantages for preterm-born adults on annual income, employment, and educational attainment, with larger disadvantages found in the extremely preterm group (24-27 gestational weeks).

Generally, it echoes most of the existing individual studies and meta-analyses of preterm birth and adulthood economic and educational outcomes. However, there is a lack of control of important family level factors (e.g., parental economic status and education levels), developmental factors (e.g., disabilities), and social factors (e.g., schooling support, community resources) to isolate the impact of preterm birth more distinctly.

Besides, below are some suggestions that might enhance the clarity and depth of the report:

Abstract

In the Results section, the authors categorized preterm births into four groups. To maintain consistency and clarity, consider using the same subgroup labels as mentioned in the Measures section: extremely preterm (24-27 weeks), very preterm (28-31 weeks), moderately preterm (32-33 weeks), and late preterm (34-36 weeks).

In the Conclusion, it is mentioned that ‘preterm birth was associated with lower economic and educational achievement in the second and third decades of life’, however, it would be more accurate to specify that the observed lower economic and educational achievements were noted in the individuals' twenties, given that the maximum age at assessment was 27/28 years, rather than extending into their thirties.

Introduction

The first paragraph provides background for challenges after preterm birth, it could be strengthened by illustrating the relevance of focusing on economic and educational outcomes. For example, need more explanation about “costs endured by families because of lifestyles and work adjustment” (Lines 139-140), and how these impact the outcomes of interest.

Lines 146-149. A more detailed explanation of the varying results of previous studies regarding preterm birth would be beneficial. Although the authors mentioned several limitations of previous studies (e.g., hospital-based tracked cohorts, small sample size, narrow range of gestation weeks), elaborating on the reasons such as differences in study design (registry vs. cohort study), measures (self-report vs. linked data), outcome indicators, and cultural contexts, would provide a stronger justification for this study.

Methods

The inclusion and exclusion criteria could be more explicitly stated. For example, 1) 1% of births with missing data on baseline covariates were excluded, what are these covariates? 2) why individuals with gestational ages below 24 weeks were excluded despite around 50% survival rate in the 1990s?

Data for employment income was obtained from ‘the T1FF records information on the personal and household income of individuals who file taxes in Canada’ (Lines 202-204). It should be clarified whether this includes only the income of the index participants or if it also includes household income. Additionally, consider addressing how individuals with low earnings that do not require tax filings are represented in the data.

For educational outcomes, why enrolment in postsecondary education was restricted to age 18-22 years? And for educational attainment, the classification is not clear, such as the last one “postgraduate degree (ISCED level: 7-8, 96 [not-applicable/medical professional training])” (Lines 227-228), not sure what this means. And whether educational attainment was treated as a binary variable with a non-university degree as the reference group or a 4-level ordinal variable. Clarification on how educational attainment was categorised in the analysis would be helpful.

Although the authors included parental demographics at birth as covariates, like maternal parity, parental age, parental birthplace, and marital status. It seems that the baseline covariates didn’t consider parental socioeconomic status or educational levels. However, cumulative evidence shows that the education and economic achievement of individuals are related to their origin family’s socioeconomic status (e.g., Sirin, 2005, Hu et al., 2022). Thus, it would be helpful to know the effect size of the contribution of preterm birth after controlling for family SES. For example, Bilsteen et al. (2022) reported that lower gestational age and lower SES at birth contributed additively to lower educational attainment across four Nordic countries.

Results

Figures 2a, 2b and 3 make it hard to differentiate these preterm subgroups due to all lines being black solid lines. This could be enhanced by using different line styles or markers for better visual representation.

In the data analysis section, the authors mention there was a subsample (n = 1,625,450 births) that linked to maternal family income at or before birth, however, the results of matching on maternal family income and other baseline characteristics were not reported in the main text, only mentioned that this “had little impact on the results (eTables 10 and 11)”. Readers might want to know more details about the differences.

Discussion

The discussion appears somewhat superficial and would be enhanced by a more thorough comparison of existing research that both supports and challenges these findings. More detailed descriptions of the data sources, methodologies, findings, and effect sizes of the studies referenced would enrich the analysis. For example, elucidating how previous studies defined and measured socioeconomic outcomes, and how these methods differ from those used in the current study, would provide valuable context. Additionally, exploring the potential reasons for the generally larger differences observed in this study compared to previous register or cohort studies would be welcomed.

The interpretation of the relative risks (RRs) for educational and income levels needs a clearer practical explanation, particularly when RRs are close to 1.0, what would this mean in plain language?

The comparative analysis between cohorts and across different periods (e.g., Swedish studies from two decades) is valuable. It would be beneficial to discuss potential reasons for observed consistent gaps or changes over time in the results, considering factors like advancements in neonatal care or shifts in societal structures. This would also help the understanding of preterm development across generations and imply targeted interventions.

The limitations section should acknowledge the potential impact of unaddressed confounding factors, such as parental educational levels, access to healthcare, and educational opportunities, which may significantly influence the outcomes observed.

6. PLOS authors have the option to publish the peer review history of their article (what does this mean?). If published, this will include your full peer review and any attached files.

Reviewer #1: No

Reviewer #2: No

---

## [Author Response · Author response to Decision Letter 0]

29 Jul 2024

Author response letter

We thank the editor and reviewers for their comments. Please see below our responses to specific comments. 

Reviewer #1: 

Our goal as a neonatologist is no longer just improved survival rate, but also intact survival, and therefore, we always think about whether PTB can become independent as a human being and contribute to society. This study results nicely demonstrated these points, and I hope the governments will promote access to educational and developmental intervention programs in early life to improve long-term outcome, with these outcomes.

However, what we must be careful about is that lives should not be chosen just because they do not contribute to the economy, as the authors mentioned in Discussion. So, I would like you intensively to emphasize this point.

Reply: We thank the reviewer for the positive comment. We have modified the implications section to emphasize the points raised above (discussion section, page 18):

“Policymaker, care providers, and parents should be aware that the socioeconomic effects of PTB is not limited to the early neonatal period but may extend into adulthood. These findings should be seen as a call to action rather than discouragement, emphasizing that all individuals, regardless of their present or future economic contributions, receive the care and support they need. By understanding the potential challenges associated with PTB, these results could inform policies that promote access to educational and developmental intervention programs in early life to improve long-term outcomes.”

l Also, as times have changed, I think there are various forms of employment and education that did not exist in the past. Moreover, this study did not include parameters solely from the family's perspective, and did not evaluate family happiness, which cannot be evaluated from an economic perspective. Of course, I know this is a scientific paper, and the content should not be spiritual or conceptual, but I think it should never go in the direction of discouraging neonatal medicine. I wanted you to keep in mind that point.

Reply: We have modified the study implications section and included these outcomes as potential future research directions (discussion section, page 19):

“Future work should also elucidate potential mediating variables for the observed associations (e.g., developmental disabilities). Additionally, it should highlight long-term impacts of PTB on social functioning and family relationships.”

Reviewer #2: Thank you for the opportunity to review this manuscript titled “Association between preterm birth and economic and educational outcomes in adulthood: A population-based matched cohort study”. This study offers valuable insights into the long-term effects of preterm birth, conducted using tax data for a large, population-based cohort of approximately 2.4 million individuals born in Canada between 1990 and 1996. The results presented indicate disadvantages for preterm-born adults on annual income, employment, and educational attainment, with larger disadvantages found in the extremely preterm group (24-27 gestational weeks).

Generally, it echoes most of the existing individual studies and meta-analyses of preterm birth and adulthood economic and educational outcomes. However, there is a lack of control of important family level factors (e.g., parental economic status and education levels), developmental factors (e.g., disabilities), and social factors (e.g., schooling support, community resources) to isolate the impact of preterm birth more distinctly.

Reply: We thank the reviewer for their comments. We have modified the limitation section to emphasize the points raised above (discussion section, page 18):

“Although we accounted for differences in baseline characteristics between preterm and term-born births using matching, residual confounding cannot be ruled out given the observational nature of the study. For example, we did not adjust for parental social factors such as education, occupation, lifestyle factors and maternal morbidities and mental health conditions, which have been linked to both the risk of PTB and worse socioeconomic outcomes. [45, 48-50]”

Additionally, we added future research directions that examine the role of the above-mentioned potential mediators (discussion section, page 19):

“Future studies that focus on identifying the role of modifiable factors that mediate or moderate these associations is needed to inform effective intervention strategies. In addition, studies should consider a longer follow-up to observe if these associations extend to mid-late adulthood and beyond. Future work should also elucidate potential mediating variables for the observed associations (e.g., developmental disabilities).” 

We have included an additional analysis in which we matched on family income quintiles at baseline. We have now included further details about these analyses and their results (methods section, page 10):

“To account for confounding by family socioeconomic status, we used a subsample that was linked to maternal tax records at the time of birth (n= 1,625,480 births) and further matched on quintiles of maternal family income at or prior to birth (based on the average maternal family income during the two years before birth) and place of residence (rural or urban) obtained from maternal tax files and compared results.”

Results discussed (results section, page 14):

“Accounting for family income at baseline, along with other baseline characteristics, had little impact on the results (annual mean income difference: -$721 CAD (-821, -622), 4% lower; RR for employment 0.98 (0.98, 0.98); RR for university enrollment 0.86 (0.84, 0.88); RR for university graduation 0.89 (0.87, 0.90)) (eTables 10 and 11).”

Abstract

In the Results section, the authors categorized preterm births into four groups. To maintain consistency and clarity, consider using the same subgroup labels as mentioned in the Measures section: extremely preterm (24-27 weeks), very preterm (28-31 weeks), moderately preterm (32-33 weeks), and late preterm (34-36 weeks).

Reply: We have modified the results section as suggested (abstract, page 3):

“Results: Of 2.4 million births, 7% were born preterm (0.3%, 0.6%, 0.8%, and 5.4% born extremely preterm (24-27 weeks), very preterm (28-31 weeks), moderately preterm (32-33 weeks), and late preterm (34-36 weeks) respectively).”

In the Conclusion, it is mentioned that ‘preterm birth was associated with lower economic and educational achievement in the second and third decades of life’, however, it would be more accurate to specify that the observed lower economic and educational achievements were noted in the individuals' twenties, given that the maximum age at assessment was 27/28 years, rather than extending into their thirties.

Reply: We modified the conclusion section to make it clearer that results were observed among young adults (abstract, page 3):

“In this population-based study, preterm birth was associated with lower economic and educational achievements at least until the late twenties.”

Introduction

The first paragraph provides background for challenges after preterm birth, it could be strengthened by illustrating the relevance of focusing on economic and educational outcomes. For example, need more explanation about “costs endured by families because of lifestyles and work adjustment” (Lines 139-140), and how these impact the outcomes of interest.

Reply: We have added more background information about the economic impact of PTB on families (background section, page 4):

“Families also face additional costs due to lifestyle and work adjustments. [8, 10, 11] For instance, parents of preterm infants often need extended leave from work, resulting in lost income and career setbacks. Families may incur higher childcare costs for specialized care and need home modifications for medical equipment or accessibility. These economic and lifestyle impacts can affect the family's financial stability and quality of life, potentially harming the child's long-term socioeconomic outcomes. [8, 10, 11]”

Lines 146-149. A more detailed explanation of the varying results of previous studies regarding preterm birth would be beneficial. Although the authors mentioned several limitations of previous studies (e.g., hospital-based tracked cohorts, small sample size, narrow range of gestation weeks), elaborating on the reasons such as differences in study design (registry vs. cohort study), measures (self-report vs. linked data), outcome indicators, and cultural contexts, would provide a stronger justification for this study.

Reply: We have added more details regarding differences between studies in terms of their study design, measures, and study population (background section, page 5):

“Most previous studies have found negative associations between PTB and markers of adulthood wealth (educational attainment, income, and employment), [15-25] but some have shown no associations with socioeconomic outcomes during adulthood. [26-28] These differences could be attributed to differences in study design (prospective [15, 20-22, 26-28] vs registry-based [16-19, 23-25]), outcome measures (self-reported [15, 20-22, 26-28] vs data linkage [16-19, 23-25]), and study population (e.g., Europe [16-19, 21, 23, 24, 28] vs North America [15, 20, 22, 25-27]).”

Methods

The inclusion and exclusion criteria could be more explicitly stated. For example, 1) 1% of births with missing data on baseline covariates were excluded, what are these covariates? 2) why individuals with gestational ages below 24 weeks were excluded despite around 50% survival rate in the 1990s?

Reply: We have added more details about the study inclusion and exclusion criteria as suggested (methods section, page 6):

“We included all live births in Canada between January 1, 1990, and December 31, 1996, identified from the VSB file. After identifying 2,729,400 eligible live births, we excluded 30,380 (1%) births with missing data on baseline covariates (GA, sex, birth plurality, and/or maternal age and parity), and 4,600 births with implausible birth weight and GA combination (birth weight for GA z score >4 SD above or below the mean). We further excluded births with GA <24 weeks’ (because of the underreporting of neonatal deaths at early GA in Canada) [29, 30] or >41 weeks’ gestation (n=2,690 and 105,270 respectively) given they were not part of the population of interest.”

Data for employment income was obtained from ‘the T1FF records information on the personal and household income of individuals who file taxes in Canada’ (Lines 202-204). It should be clarified whether this includes only the income of the index participants or if it also includes household income. Additionally, consider addressing how individuals with low earnings that do not require tax filings are represented in the data.

Reply: We have clarified that employment income includes only the income of the index participant (methods section, page 7):

“Employment income per year after an individual turns 18 years and until the end of follow-up was obtained from Statistics Canada income tax family files (T1FF). The T1FF records information on the personal income of individuals who file taxes in Canada (up to 8 months after the filing deadline of April 30th). We defined the total employment income as the annual individual pre-tax sum of paid employment income (wages, salaries, and commissions), net-self-employment income, and employment insurance income.”

We additionally discussed the fact that some individuals, especially those with low income, might not file their tax returns (discussion section, page 17):

“Despite several incentives for low-income individuals to file their tax returns (e.g., social benefits), some may not meet the administrative deadline (within 8 months from the April 30th filing deadline), resulting in missing income data.”

For educational outcomes, why enrolment in postsecondary education was restricted to age 18-22 years? And for educational attainment, the classification is not clear, such as the last one “postgraduate degree (ISCED level: 7-8, 96 [not-applicable/medical professional training])” (Lines 227-228), not sure what this means. And whether educational attainment was treated as a binary variable with a non-university degree as the reference group or a 4-level ordinal variable. Clarification on how educational attainment was categorised in the analysis would be helpful.

Reply: We have added further details about how we categorized educational outcomes (methods section, page 8):

“Post-secondary Education Enrollment: We determined individuals’ enrollment in postsecondary education at age 18-22 years and categorized them according to the highest level of their program of enrollment as not enrolled, enrolled in college, or enrolled in a university program. We chose to restrict to age 18-22 years because most individuals are expected to enroll in a postsecondary program within this age range. 

Postsecondary Education Attainment: We ascertained individuals’ highest level of educational attainment by the end of follow-up (maximum age range 22-27 years) and categorized as a multinomial outcome: “did not graduate from any postsecondary education” (the reference) or “graduated with a non-university degree (International Standard Classification of Education (ISCED) level: 3-5)”, “university degree (ISCED level: 6)”, or “postgraduate degree/medical professional training (ISCED level: 7-8, 96)”.”

Although the authors included parental demographics at birth as covariates, like maternal parity, parental age, parental birthplace, and marital status. It seems that the baseline covariates didn’t consider parental socioeconomic status or educational levels. However, cumulative evidence shows that the education and economic achievement of individuals are related to their origin family’s socioeconomic status (e.g., Sirin, 2005, Hu et al., 2022). Thus, it would be helpful to know the effect size of the contribution of preterm birth after controlling for family SES. For example, Bilsteen et al. (2022) reported that lower gestational age and lower SES at birth contributed additively to lower educational attainment across four Nordic countries.

Reply: We have included an additional analysis in which we matched on family income quintiles at baseline. We have now included further details about these analyses and their results (methods section, page 10):

“To account for confounding by family socioeconomic status, we used a subsample that was linked to maternal tax records at the time of birth (n= 1,625,480 births) and further matched on quintiles of maternal family income at or prior to birth (based on the average maternal family income during the two years before birth) and place of residence (rural or urban) obtained from maternal tax files and compared results.”

Results discussed (results section, page 14):

“Accounting for family income at baseline, along with other baseline characteristics, had little impact on the results (annual mean income difference: -$721 CAD (-821, -622), 4% lower; RR for employment 0.98 (0.98, 0.98); RR for university enrollment 0.86 (0.84, 0.88); RR for university graduation 0.89 (0.87, 0.90)) (eTables 10 and 11).”

Additionally, we discussed the potential for residual confounding by these factors in the limitation section and added the suggested references (discussion section, page 18):

“Although we accounted for differences in baseline characteristics between preterm and term-born births using matching, residual confounding cannot be ruled out given the observational nature of the study. For example, we did not adjust for parental social factors such as education, occupation, lifestyle factors, and maternal morbidities and mental health conditions, which have been linked to both the risk of PTB and worse socioeconomic outcomes. [45, 48-50]”

Results

Figures 2a, 2b and 3 make it hard to differentiate these preterm subgroups due to all lines being black solid lines. This could be enhanced by using different line styles or markers for better visual representation.

Reply: We have now uploaded colored figures as TIFF files. 

In the data analysis section, the authors mention there was a subsample (n = 

---

## [Decision Letter · Decision Letter 1]

25 Sep 2024

Association between preterm birth and economic and educational outcomes in adulthood: A population-based matched cohort study

PONE-D-23-38655R1

Dear Dr. Pechlivanoglou,

We’re pleased to inform you that your manuscript has been judged scientifically suitable for publication and will be formally accepted for publication once it meets all outstanding technical requirements.

Kind regards,

Quetzal A. Class, PhD

Academic Editor

PLOS ONE

Additional Editor Comments (optional):

Thank you for being so responsive to the reviewer suggestions. Again, apologies for the lengthy review process. Congratulations!

Reviewers' comments:

Reviewer's Responses to Questions

**Comments to the Author**

1. If the authors have adequately addressed your comments raised in a previous round of review and you feel that this manuscript is now acceptable for publication, you may indicate that here to bypass the “Comments to the Author” section, enter your conflict of interest statement in the “Confidential to Editor” section, and submit your "Accept" recommendation.

Reviewer #2: All comments have been addressed

2. Is the manuscript technically sound, and do the data support the conclusions?

Reviewer #2: Yes

3. Has the statistical analysis been performed appropriately and rigorously? 

Reviewer #2: Yes

4. Have the authors made all data underlying the findings in their manuscript fully available?

Reviewer #2: Yes

5. Is the manuscript presented in an intelligible fashion and written in standard English?

Reviewer #2: Yes

6. Review Comments to the Author

Reviewer #2: The authors have been very responsive to each of this reviewer's comments. The figures are now clearer using colour.

7. PLOS authors have the option to publish the peer review history of their article (what does this mean?). If published, this will include your full peer review and any attached files.

Reviewer #2: **Yes: **Dieter Wolke

---

## [Editor Report · Acceptance letter]

3 Oct 2024

PONE-D-23-38655R1 

PLOS ONE

Dear Dr. Pechlivanoglou, 

I'm pleased to inform you that your manuscript has been deemed suitable for publication in PLOS ONE. Congratulations! Your manuscript is now being handed over to our production team.

Kind regards, 

on behalf of

Dr. Quetzal A. Class 

Academic Editor

PLOS ONE